# Farmers' Preferences for Cotton Cultivation Characteristics: A Discrete Choice Experiment in Burkina Faso

**Edouard I. R. Sanou** [1]**, Juan Tur-Cardona** [1]**, Jeffrey D. Vitale** [2]**, Bazoumana Koulibaly** [3]**, Godelieve Gheysen** [4]  **and Stijn Speelman** [1],*

1. Department of Agricultural Economics, Ghent University, 9000 Gent, Belgium; rachidjunior@yahoo.fr (E.I.R.S.); juan.turcardona@ugent.be (J.T.-C.)
2. Department of Agricultural Economics, Oklahoma State University, Stillwater, OK 74078, USA; jeffrey.vitale@okstate.edu
3. Programme Coton, Institut National de l'Environnement et de la Recherche Agricole (INERA), Ouagadougou BP 7044, Burkina Faso; bazoumana@hotmail.com
4. Department of Biotechnology, Ghent University, 9000 Gent, Belgium; Godelieve.Gheysen@UGent.be
* Correspondence: stijn.speelman@ugent.be; Tel.: +32-(0)92649375

**Abstract:** While a fierce debate about the advantages and disadvantages of genetically modified crops is ongoing, it is surprising that farmers are often not consulted. In Burkina Faso, where insect resistant Bollgard II® cotton (further termed Bt cotton) was commercially released in 2008, studies highlight that cotton producers are in general satisfied with the reduction in insecticide use while the economic benefits are a source of controversy. To gain insight into farmers' preferences towards attributes in cotton cultivation, a discrete choice experiment (DCE) was developed. Five key attributes were identified to describe improved cotton varieties: seed development and provenance, seed costs, yield, required number of insecticide sprays, and preservation of agricultural practices. Farm-gate surveys were conducted among 324 cotton farmers in Western Burkina Faso. The results show that overall, farmers have a positive preference towards yield improvements and a negative preference towards pure private seed development and towards an increase in the requested number of insecticide applications or in the seed costs. According to their varieties at the time of the surveys (Bt and non-Bt), a difference was observed regarding their preferences for a status quo situation, indicating that those growing Bt had a stronger preference to keep the status quo than non-Bt farmers. When dividing the sample in segments based on the farm size, it was shown that there were different preferences with respect to the development of the variety and the required number of insecticide applications. Overall, it can be concluded from this study that economic benefits (linked to higher yields, lower seed costs, or reduced pesticide use) shape farmer's preferences.

**Keywords:** GMOs; Bollgard II®; Bt cotton; discrete choice experiment; conditional logit model; farmers; Burkina Faso

---

## 1. Introduction

The adoption of genetically modified (GM) crops in Sub-Sahara Africa (SSA) is highly controversial [1,2]. Although several authors claim that agricultural biotechnology presents an opportunity to alleviate starvation and poverty, malnutrition, and food insecurity [3–6], the adoption of GM crops is lagging behind in SSA compared to other parts of the world [7–9].

Burkina Faso was an exception to this. Already, in 2008, Bollgard II® cotton was commercially released in Burkina Faso. Evidence of the first five years of Bt-cotton commercialization in Burkina Faso

positioned its positive experience as a roadmap for eventual wider adoption of GMOs in Africa [10] and as a success model towards sustainable agriculture practices [11]. Also, the Food and Agriculture Organization (FAO) [12] followed the experiences of Burkina Faso with interest due to the particular dominating position of smallholders in the agricultural sector. However, in 2015, because of an issue with the length of the cotton fibers, the cotton companies convinced the government of Burkina Faso to suspend the cultivation of Bt-cotton [13]. At this moment, the quality issue discussion is still ongoing, with the perspective to fix the backcross issue pointed as non-sufficient to ensure the carry-over of the desired beneficial traits [14].

At the moment of the phase-out, most studies on the commercialization of Bollgard II$^{®}$ in the Burkina Faso farming system had been largely positive about the outcomes [8,10,15,16]. The majority of these studies, whether using ex-ante [9,17,18] or ex-post approaches [15,18,19], focused on the overall economic benefits based on the overall yield gains and the production costs compared with conventional cotton. However, the debate on the potential economic benefits of the GM technology for small farmers' welfare is still undecided [20–22] and several authors have indicated that the higher seed cost of Bt-cotton specifically affected poor farmers [18,19].

Recently, Sanou et al. [23] investigated farmers' opinions towards Bt cotton in Burkina Faso, pointing out that small or subsistence farmers (who dominate the cotton sector of Burkina Faso) were not entirely positive about the economic benefits of Bollgard II$^{®}$, mainly because of the seed pricing policy. These results seem to support the concern that Bollgard II$^{®}$ might not entirely match with farmers' expectations. In general, it has also been noted by some authors, e.g., Carro- Ripalda and Astier [24], that while smallholder producers are the ones most likely to be affected by the introduction of GM crops, they are the least included in public debates and consultation about the development, implementation, or regulation of agricultural biotechnology. Also, in Burkina Faso, where the cotton value chain is organized in a very top-down manner, farmers' involvement in the development and spread of the crop has been limited. The variety was developed in a public private partnership between INERA (Institut National de l'Environnement et de la Recherche Agricole) and Monsanto and essentially the cotton companies decide which varieties the farmers grow.

In this context, it is particularly relevant to focus on farmers' preferences for the key attributes of improved cotton varieties, because this could steer future developments. A discrete choice experiment (DCE) considering different hypothetical GM varieties was used to assess which traits farmers would value more. According to Breustedt et al. [25], there is relatively little empirical evidence in the literature on farmers' preferences for GM technology using a DCE. Some exceptions are [26–29]. Additionally, to the best of our knowledge, apart from the recent ex-ante study by Chinedu et al. [30], studies in SSA have been even rarer.

This paper is organized in three sections. First, the approach as well as the method adopted to perform the DCE is explained. Second, the outcomes from the DCE analysis are presented and discussed. Finally, in the last section, some conclusions are made.

## 2. Discrete Choice Experiment in Cotton Cultivation: Approach and Method

### 2.1. Cotton Cultivation and Agricultural Biotechnology in Burkina Faso

Cotton is the most important cash crop in Burkina Faso. In 2014, it was estimated that the cotton sector not only provided labor for more than 350,000 farmers but also indirectly contributed to the livelihood of more than 3 million people, taking into account the entire value chain and that of by-products, such as local oil factories and cattle food producers [31]. The importance of the cotton sector for Burkina Faso was confirmed in a background paper to the United Nations Conference on Trade And Development–Food and Agriculture Organization (UNCTAD-FAO) Commodities and Development Report 2017 [32]. The cotton production chain in Burkina Faso is administrated through vertical integration between farmers and cotton companies. There are only three cotton companies: two private companies (Socoma and Faso Coton) and one parastatal (Sofitex), who each control a

specific area. Of these, Sofitex is the largest, covering more than 85% of the land cultivated with cotton and representing about 80% of the national cotton production [31]. At the village level, cotton producers are organized in producer groups or GPC (Groupement des Producteurs de Coton) and the cotton companies provide inputs, such as seeds, pesticides, fertilizers, and technical advice, to GPC, who later sell their cotton to the company at a guaranteed price [33–35]. In this system, it is the cotton company that decides which variety farmers grow.

### 2.2. Sampling Method

Given that an individuals' perception of the risk and benefits of a new technology is determined by selected sources of information, values, interests, and individual experiences [36], our surveys were conducted with farm decision makers at the farm gate. A total of 324 farmers were interviewed during the 2015–2016 agricultural season (November–January) across three regions in the west of Burkina Faso, an area administered by Sofitex. Within this zone, three districts (Dedougou-Bobo-Diebougou) were selected along a north–south gradient presenting different agro climatic conditions. Dedougou receives 700 to 900 mm of precipitation per year, Bobo 800 to 1000 mm, and Diebougou 1000 to 1100 mm. Accounting for agro-climatic conditions is important because they are known to be determinant factors influencing cotton production in Burkina Faso [19]. Moreover, the three districts considered host 7 of the 13 Sofitex factories so they can be considered important cotton-producing districts. From each district, four villages were randomly selected. At the village level, the list of cotton farmers provided by Sofitex was used as a master list.

A stratified sampling method was adopted in the selection of farmers from the master list. This stratification relied on two key features: Type of variety currently grown and type of farmer. Farmers growing conventional or Bt varieties have different yields and experience, and will have to implement different farming practices that might influence preferences. Furthermore, as highlighted in the introduction, the benefits of the varieties might not be the same for all farmers. Therefore, we decided to look at the type of farmer according to the size, using categories previously defined by INERA (small, medium, and large). Within each stratum, similar numbers of farmers were randomly selected.

The survey team consisted of the first author and five students recently graduated from the Rural Development Institute of the Polytechnic University of Bobo Dioulasso. The students were selected based on their previous survey experiences and their knowledge of local languages (Moore, Bobo, Dioula, or Dagara) to enable coverage of the predominant languages in each district. Before the interviews were initiated, the enumerators were trained and exposed to the objectives of the study to ensure consistency in the execution of the survey and explanation of all concepts.

### 2.3. Discrete Choice Approach

Choice experiments (CEs) have been widely used in the agricultural and environmental economics literature and their use in development economics is rising [37,38]. Several studies have used CEs to evaluate farmers' behavior and preferences [37,39,40]. An advantage of using CE is that it is a technique for eliciting preferences to understand farmers' demands for new varieties where it is impossible to use revealed preference data on the actual choices made by farmers. In our study, the CE performed embodies the adoption of a GM crop. The crop considered was an insect resistant cotton variety. Farmers chose between different alternatives involving GM variety attributes, according to their own preferences and budget constraints. The data from the CE were analyzed using a conditional logit (CL) model.

### 2.4. Econometrical Model

The conditional logit model originates from random utility theory. A farmer i faces J choices, including various alternatives or keeping his status quo. Suppose that the utility level of choosing alternative j for this farmer is:

$$U_{ij} = \beta' X_{ij} + \varepsilon_{ij} \tag{1}$$

where $X_{ij}$ is a vector of choice-specific attributes. For the conditional logit, the parameter β is constant across choices. A farmer chooses scenario j, if the utility, $U_{ij}$, is the highest among all J choices (i.e., $U_{ij} > U_{ik}$ for all k ≠ j). To extend the CL model, an additional error, $\varepsilon_{ij}$, is incorporated into the model to capture any remaining status quo (SQ) effects in the stochastic part of utility [41]. Thus, when a choice j is made, the statistical model for the probability between alternatives k and j can be represented as:

$$P(yi = j) = P(Uij > Uik) \; \forall \; k \neq j. \tag{2}$$

Following Bonnichsen and Ladenburg [42], the addition of an alternative specific constant (ASC) for the SQ, reduces the status quo bias and improves the internal validity of the stated preferences. In principle, the respondents thus only make trade-offs with regards to the attributes, and potential status quo effects are cancelled out as it enters the utility function for the entire alternative [43]. The observable component, $X_{ij}$, is assumed to be a linear function:

$$X_{ij} = ASC + \alpha_m Y_{mj}, \tag{3}$$

where α denotes a vector of preference parameters associated with attribute m, $Y_{mj}$ is a vector of attributes of alternative j, and ASC denotes an alternative specific constant. The ASC for the SQ alternative is a dummy variable that controls for the utility associated with the SQ alternative relative to the hypothetical alternatives in choice experiments [43]. Moreover, Scarpa et al. [41] argue that including an ASC in the econometric analysis captures unobservable influences beyond attributes present in the choice sets. A significant ASC, representing the SQ alternative or the designed choice alternatives, then might suggest that a status quo effect occurs [44,45]. Additionally, given that the farmers in the sample were growing different varieties in the past, the ASCs interact with the variety that they are growing (Bt and non-Bt). Thus, the attitude towards the ASC for the different growers will be captured.

In addition, by including a monetary attribute (seed cost), it was possible to estimate the willingness to pay (WTP) for the non-seed cost attributes. This was done by dividing the non-price attribute with the seed price attribute, as specified below:

$$WTP = -\beta_x/\beta_{price}, \tag{4}$$

where $\beta_x$ is the coefficient of the attribute of interest and $\beta_{price}$ is the price coefficient.

*2.5. Identification of Attributes and Levels*

As in all DCE applications, the identification of the attributes and levels is a first important step. Relevant attributes were identified by combining a literature review with experts' opinion. The literature included previous studies on the adoption of Bt cotton in Burkina Faso, as well as studies on farmers' perception and adoption decisions of transgenic crops in Sub-Saharan Africa (SSA) [46–48]. The engaged experts included researchers from INERA and the Ministry of Agriculture and Food Security in Burkina Faso. The experts were selected based on their knowledge of farming systems in Burkina Faso. During the expert consultation, the final attributes were selected. Often, focus groups are organized to define an appropriate set of attributes, but because of the security situation in the country at the time of the design, individual expert consultations were chosen. Finally, pretesting of the CE with some farmers confirmed that the attributes included were relevant.

The varieties were described using five attributes (Table 1).

**Table 1.** Attributes and levels of attributes for the choice experiment.

| | | Levels of Attribute | | | |
|---|---|---|---|---|---|
| **Attributes** | | **1** | **2** | **3** | **4** |
| 1. Number of Sprays (Treatment) | | 6T | 4T | 2T | |
| 2. Seed Price (FCFA [1]) | | 30,000 | 25,000 | 17,500 | 10,000 |
| 3. Seed Provenance | | Private | Public | Public-private partnership [*] | |
| 4. Agricultural Practices | | Change | No change [*] | | |
| 5. Yield (kg/ha) | Small (≤2ha) | 675 | 750 | 900 | |
| | Medium (<2ha–5ha≤) | 900 | 1000 | 1200 | |
| | Large (>5ha) | 1200 | 1350 | 1600 | |

[1] FCFA: franc de la Communauté financière africaine, 1 FCFA = 0.0017$ at time of interviews. [*] Represent the Status Quo level, for "Yield" and "Number of Spray" the actual individual values reported by the farmers are inserted as Status Quo level for the modeling. "Seed price" the Status Quo level is the current price on the market of 27,000 FCFA for Bt producers and 7000 FCFA for conventional producers ($1 = 592 FCFA).

*Required number of insecticide treatments:* The recommended number of treatments in Burkina Faso for conventional cotton is six. The first four treatments target Lepidopteran insects and the last two treatments control secondary pests, such as jassids and aphids. The existing GM variety allows farmers to spray only twice, targeting the secondary insect groups. However, a third level of four insecticide treatments was defined to see whether there is a tradeoff with other characteristics. This third level is also sustained by the trends in Bt cotton farms, where an increased abundance of non-targeted insects was observed, so that the number of required insecticide treatment tends to increase [23].

*Seed price:* The existing GM variety grown in Burkina Faso costs 27,000 FCFA per sack and one sack is supposed to be the quantity of seeds for one ha. However, the seed price has been pointed at by a number of studies as the main constraint impacting economic benefits. The high price is even more problematic because changing climatic conditions have led to the situation that farmers often have to buy extra seeds, in that way spending even more than 27,000 FCFA per ha. To appraise which price will be acceptable for farmers, three lower price levels (25,000; 17,500; 10,000 FCFA) and one higher price of 30,000 FCFA were used.

*Seed development:* Seed provenance has been at the core of the GMO debate, where the opponents criticize the monopolistic way multinationals control seed production. In the case of Burkina Faso, however, the Bollgard II® seeds were developed under a partnership between the lead cotton company, Sofitex, and Monsanto. In our experiment, three levels were included: public, public private partnership, and private.

*Change in agricultural practices:* The cultivation of GM crops might also require changes in agricultural practices. This attribute captured whether such changes are necessary or not.

*Yield:* To obtain a reliable range of yields, the Sofitex database on 10 years of production was consulted. This covered the last 5 years before and the first 5 year after Bollgard II®'s commercial release. Finally, three levels of yield were used and adjusted based on the type of farmer (large, medium, and small).

To sum up, a total of three attributes with three levels ($3^3$) were coupled to an attribute of four level (4) and two levels (2). In the analysis, effects coding was used for the categorical attributes.

*2.6. Design of Choice Sets*

Once the attributes and their levels are identified, an appropriate design should be constructed. This involves combining the attribute levels into choice profiles (or alternatives) and grouping the profiles in choice cards [49]. Thus, according to the five identified attributes and their levels (Table 2), a fractional factorial design generates a sample of the full design ($33 \times 4 \times 2 = 264$) in such a way that the most important effects are estimated [50]. The D-efficiency approach was used to design the experiment with the help of SAS software [51]. A D-efficient design tends to greatly reduce the predicted standard errors of the parameter estimate and produce even stronger statistical results [52,53].

**Table 2.** An example of the choice set addressed to the small farmers group.

| Attributes | Alternative 1 | Alternative 2 | Alternative 3 |
|---|---|---|---|
| Yield (kg/ha) | 675 | 900 | 750 |
| Number of Sprays | 4 T | 2 T | 6 T |
| Seed cost (FCFA) | 17,500 | 10,000 | 25,000 |
| Seed provenance | Public private partnership | Private | Public |
| Agricultural practices | Change | Change | No Change |

**Alternative 4**: I prefer to maintain my current way to grow cotton
**Alternative 5**: I would like to stop growing cotton

Two opt-out alternatives were included (Table 2). The first one refers to the baseline, meaning farmers could choose to continue their current way of growing cotton, while the second opt-out allowed farmers to indicate a preference for stopping growing cotton when compared to the other alternatives. Following Veldwijk et al. [54], including an opt-out option in DCE leads to an unforced choice model, which may therefore induce a downward bias [55]. The choice sets were compiled by means of SAS (Statistical Analysis System software). In total, 24 cards containing 3 alternatives were generated. In the survey, each farmer was confronted with 8 randomly chosen choice sets. This was done by splitting the 24 choice sets generated by SAS into 3 blocks of 8 choice sets. Blocking helps to promote response efficiency by reducing cognitive effort for each respondent [56]. A total of 12,960 individual choices were obtained by the study (5 alternatives × 8 choice cards × 324 farmers). Table 2 below presents an example of one of the choice cards.

## 3. Results and Discussion

### 3.1. Farmer Characteristics

All of the farmers surveyed were male and the majority were in the age groups 31–40 (39.2%) and 41–50 (43.4%). Most farmers (over 80%) had not had any formal education. Only 2.8% of the selected farmers had received formal education in agriculture, 11.4% had followed primary education, and 4.9% secondary. The overall experience of the selected farmers with cotton production was slightly greater than 25 years. Seven years after the introduction of Bollgard II® in Burkina Faso, the farmers growing Bt had, on average, 6 years of experience with this crop, but even those not growing Bt at the time of the survey had on average tried it for 2 years. This is linked to the top-down decision made by Sofitex about the variety distributed to each GPC, the surface under Bt cotton gradually increased to 70% over time, but the Bt variety nor the traditional variety was given systematically to GPCs. The average surface cultivated with cotton by the interviewed farmers was around 4.5ha. When looking at the yield performance (Table 3), Bt growers on average had a yield of 1,115.27 kg/ha and non-Bt growers on average 953.23 kg/ha.

**Table 3.** Basic demographic background and yield component from the farmers surveyed.

| Parameters | Type of Farmers | | | | |
|---|---|---|---|---|---|
| | Small (*n* = 108) | Medium (*n* = 108) | Large (*n* = 108) | Total (*n* = 324) | |
| **Age** | | | | | |
| up to 20 | 2 | | | 2 | **0.6%** |
| 21–30 | 9 | 11 | 2 | 22 | **6.8%** |
| 31–40 | 37 | 55 | 35 | 127 | **39.4%** |
| 41–50 | 52 | 27 | 62 | 141 | **43.4%** |
| 51–60 | 7 | 13 | 7 | 27 | **8.4%** |
| over 60 | 1 | 2 | 2 | 5 | **1.6%** |
| **Education level** | | | | | |
| Non | 29 | 35 | 39 | 103 | **31.8%** |
| Non formal | 56 | 51 | 52 | 159 | **49.1%** |
| Primary | 13 | 11 | 13 | 37 | **11.4%** |
| Secondary | 7 | 7 | 2 | 16 | **4.9%** |
| Formal agricultural background | 3 | 4 | 2 | 9 | **2.8%** |
| **Yield Component (2015–2016 campaign)** | | | | | |
| Average surface (ha) | 1.7 | 3.5 | 7.9 | **4.4** | |
| Yield (kg/ha) for Bt growers (*n* = 162) | 989.72 | 1121.64 | 1234.64 | **1115.27** | |
| Yield (Kg/ha) for Non-Bt growers (*n* = 162) | 870.7 | 960.74 | 1028.24 | **953.23** | |
| Yield improvement (Bt vs. Non-Bt) | 12% | 14.3% | 16.7% | **14.5%** | |

*3.2. Conditional Logit*

The CL model was estimated using NLOGIT 5 software. Two CL models were performed, with both considering general farmers' preference of the entire sample as well as the preferences according to the farm size parameter. The estimation results of the conditional logit model for the whole sample (Table 4) showed that all of the attributes are significant determinants of farmers' preferences. Farmers in our sample had a preference for higher yields and a lower number of required insecticide treatments. The preference towards the number of sprays supports previous studies indicating that insect resistance is one of the main reasons spurring the adoption of Bt cotton in Burkina Faso [57,58]. Cotton growers in our sample also preferred seeds to be developed under a pure public or a public-private partnership above a pure private initiative and they preferred affordable prices. Increases of the current seed price would be negatively perceived by the farmers in our sample. The current collaborative partnership was found to be suitable by farmers. A similar adhesion of farmers to this parastatal seed development system was previously mentioned in a study conducted on the assessment of the impact of institutional factors on Bt cotton implementation [19,33]. Considering the agricultural practices attribute overall, farmers of this study, when opting for an improved variety, expressed a positive preference to change their current way to grow cotton.

**Table 4.** Conditional logit model outcomes for the total sample.

| Utility Parameter | Coefficient | Standard Error |
|---|---|---|
| Yield | 0.00418 *** | 0.00013 |
| Private seed source [1] | −1.10502 *** | 0.06893 |
| Public seed source [1] | 0.77039 *** | 0.04905 |
| Number of spray | −0.59463 *** | 0.02548 |
| Seed price | −0.08727 *** | 0.00492 |
| Agricultural practices | 0.13576 *** | 0.03719 |
| $ASC_{SQ\ x\ Bt\ grower}$ | 0.42554 *** | 0.06119 |
| $ASC_{SQ\ x\ NonBt\ grower}$ | −0.08359 * | 0.04531 |
| **Probability of selection an opt out alternative** | | |
| Alt4: Preference for the status quo | 51% (58.6% Bt, 43.2 Non-Bt) | |
| Alt5: Abandon of cotton growing | 0% | |

[1] Compared to public-private partnership; $ASC_{SQ}$: Alternative Specific Constant for the Status quo; ***, * = Significant level at 1%, 10% level.

The rate of choosing the status quo variety was also valued compared to the other alternative varieties. Table 4 also provides information about the share of farmers preferring the status quo or the choice of abandoning cotton cultivation. None of the farmers indicated a preference to stop growing cotton. Our model showed that the Bt and non-Bt growers had an opposite preference towards their current situation. When considering the entire sample, about 51% of the participants preferred the status quo. However, the segmentation results based on the type of variety grown revealed that nearly 59% of Bt farmers preferred the status quo versus slightly more than 43% of the conventional growers. So, while a small majority of Bt growers had a tendency to choose the status quo variety, a more negative attitude towards the current variety was observed from the preferences of non-Bt farmers. This confirms earlier studies [8,10,15,16] that found that farmers are quite satisfied with the characteristics of the current Bollgard II® variety. Considering the non-Bt growers, approximatively 60% in our sample expressed a willingness to adopt an alternative GM variety with a higher yield and low number of insecticide treatments at an affordable price.

When looking at the results of the conditional logit for different farm sizes (Table 5), it is interesting to consider that slightly more than half of the "small" and "large" farmers groups preferred the status quo. The negative attitude towards the ASC for small and medium non-Bt growers suggests that these farmers might not be satisfied with these crops for reasons beyond the described attributes. However, the results also highlighted that Bt farmers cultivating less than 2 ha are also not satisfied with the Bollgard II® variety. This level of appreciation from small-scale farmers could be related to the findings of Renaudin et al. [18], who highlighted the financial risks for small farmers attributed to Bt cotton cultivation owing to the high seed price of Bt cotton. More recently, Sanou et al. [23] also found that the seed price policy was the main constraint listed by small-scale farmers.

**Table 5.** Conditional logit model estimation based on the type of farmers.

| Utility Parameter | Small | Medium | Large |
|---|---|---|---|
| Yield | 0.00721 *** (0.00036) | 0.00537 *** (0.00027) | 0.00499 *** (0.00024) |
| Private seed source [1] | −1.30542 *** (0.13210) | −1.36135 *** (0.12297) | −1.59307 *** (0.13728) |
| Public seed source [1] | 0.81443 *** (0.09442) | 0.80564 *** (0.08351) | 0.91111 *** (0.09185) |
| Number of Spray | −0.76542 *** (0.05320) | −0.72264 *** (0.04782) | −0.91640 *** (0.05623) |
| Seed Price | −0.10467 *** (0.00947) | −0.10353 *** (0.00864) | −0.12168 *** (0.00970) |
| Agricultural practices | −0.13919 ** (0.07015) | 0.27694 *** (0.06355) | 0.27705 *** (0.06874) |
| ASCSQ × Bt grower | −0.53640 *** (0.13005) | 0.09291 (0.10713) | 0.82464 *** (0.10766) |
| ASCSQ × NonBt grower | −0.79085 ***(0.09439) | −0.23509 ***(0.07887) | 0.37331 ***(0.08366) |

| Probability of selection an opt out alternative | |
|---|---|
| Alt4: Preference for the status quo | 54.9% (Small), 46.4% (Medium), 51.4% (Large) |
| Alt5: Abandon cotton growing | 0% for all |

[1] Compared to public-private partnership; $ASC_{SQ}$: Alternative Specific Constant for the Status quo; ***, **, = Significant level at 1%, 5%, level, (...) = Standard Error.

When considering the ASC for the large farmers groups, our model seems to suggest that they were satisfied with their current situation regardless of whether they were growing the Bt or non-Bt variety. The assessment of large farmers' preferences in this model revealed that about 51.4% adhered to the status quo variety.

The willingness to pay (WTP) for the non-seed cost attributes changes was estimated and reported in Table 6. Considering yield gains, our data suggest that farmers are willing to pay an average of 48 FCFA to increase their cotton yield with one-unit kilogram. However, when comparing the WTPs of each farmer group, it is interesting that the WTP of the "small" farmers group was the highest (69 FCFA/kg), followed by the medium (52 FCFA/kg) and large (41 FCFA/kg) farmers groups. A reason for this might be the low yield performance faced by small-scale farmers.

**Table 6.** Willingness to Pay (WTP) for attribute level changes.

| | Willingness to Pay | | | |
| --- | --- | --- | --- | --- |
| | Farmers | Farm Size | | |
| Attributes | All | Small | Medium | Large |
| Extra Yield (FCFA/kg) | 48 | 69 | 52 | 41 |
| From PP partnership to Private seed source (FCFA) | −16,496 | −17,163 | −18,517 | −18,697 |
| Public seed source (FCFA) | 4993 | 3090 | 2414 | 1883 |
| Extra insecticides treatment (FCFA/Treatment) | −6814 | −7313 | −6980 | −7531 |
| Preparedness to change their Agricultural practices | 3111 | −2659 | 5350 | 4554 |

PP: Public-Private.

The change from private to public-private partnership seed development presented the highest WTP value (an average of 16,496 FCFA) when compared to the other attributes. There was no significant difference among the farmer groups. This confirms that our sample farmers are willing to pay more to stay in the public-private partnership compared to moving to a pure private seed source. Similarly, the WTPs for eliminating one insecticide treatment were not significantly different among the farmer groups. This also sustains the aforesaid findings, indicating that all sample farmers have a clear preference for a cotton variety that requires a low number of spraying.

Regarding farmers' preparedness to change their current agricultural practices, it is noteworthy to see that only small farmer groups were more reluctant to change their current practice. A reason for that might be that they are more concerned about improving their current yield than taking a risk to go for a change that can increase the production cost. Both the large and medium farmers groups presented a non-significant WTP value.

## 4. Conclusions

Until 2016, the introduction of Bt cotton in Burkina Faso was generally regarded as a huge success [8,10,11]. This perception was mainly based on the fast uptake and evaluations of the average productivity gains. Farmers' opinions were seldom heard in this. Furthermore, critical voices have also criticized the top-down development and introduction of the crop and the high level of vertical integration in the cotton supply chain, where famers have to grow the seeds they receive from the cotton companies. In light of this, the objective of this study was to investigate whether the available variety matches with the preferences of farmers and which direction crop development should take. This study therefore analyzed preferences by farmers for attributes of hypothetical GM cotton varieties in western Burkina Faso. A similar methodology could also be used before GM crops are introduced in a country. In this way, it assessed whether particular crop varieties meet existing needs.

Outputs of this study confirmed that all of the identified attributes appeared to be significant determinants of the preferences of farmers. It was shown that farmers have a positive preference towards yield improvements and a negative preference towards pure private variety development and towards an increase in the required number of insecticide applications as well as towards an increase in the seed costs. Overall, the development of new seed varieties whether by public institutions or public-private partnership was positively perceived by farmers. The majority of the farmers involved in this study are also open to changing their agricultural practices. This is important because the earlier study of Ezezika et al. [59] identified change in traditional agricultural practices as an important influencing factor to consider in order to guarantee the success of the implementation of biotechnology crops in SSA.

When looking at the proportion of preferences for the "status quo" option, the study found that nearly 59% of Bt growers in our sample were satisfied with their current situation. On the other hand, 60% of farmers growing conventional cotton were willing to adopt one of the hypothetical GM varieties. When comparing different farm size groups, about 55% of the small and 51% of the large farmers chose the status quo.

Considering the WTP estimates for attribute changes, the highest value was found for the attribute linked to public-private partnership seed sources. The WTP for eliminating an insecticide treatment was not significantly different among farmer groups. Small farmers were found to dislike changes in their agricultural practices. To conclude, this study found that farmers' preferences are mainly shaped by the economic benefits linked to higher yields and the reduction of the number of sprays and seed cost. The current public/private partnership was found to be an attractive option from a farmer's point of view. Also, public seed development could be accepted by farmers while pure private initiatives ware badly perceived. Our findings confirm the important role farmers see for the government in terms of crop development. They also show that while appreciation for the Bt cotton variety, which was cultivated in Burkina Faso up to 2016, overall seems quite good, it is relevant in crop development to take into account the observed differences between farm groups.

**Author Contributions:** The conceptualization and methodology of the study was realized and approved by E.I.R.S., G.G., S.S.; investigation and formal data analysis, E.I.R.S., J.T.-C., S.S.; writing—original draft preparation was done by E.I.R.S. and S.S. writing—review and editing was interactively realized by E.I.R.S., G.G., S.S., J.T.-C., J.D.V. and B.K.; supervision S.S.; project administration, S.S. and G.G.; funding acquisition G.G. and S.S.

**Funding:** This work was funded by the ERAfrica FWO-project GA.013.14N.

**Conflicts of Interest:** The authors declare no conflict of interest.

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
