# Peer review of "Farmers’ Preferences for Cotton Cultivation Characteristics: A Discrete Choice Experiment in Burkina Faso"

_agronomy, doi:10.3390/agronomy9120841_

Round 1
Reviewer 1 Report
Once again I would like to congratulate the authors for a sound and relevant paper that helps explain the Bt cotton situation in Burkina Faso. There has been a lot of speculation and even some misrepresentation about what happened in Burkina with regard to the Bt cotton story. I know this paper will contribute to clear up the story behind the technology in the country.
Hopefully, this paper will ellicit more scholarly analysis that will help understand the pitfalls and issues related to technology adoption in Burkina and the region. Deploying potentially valuable technologies without addressing institutional and organizational constraints can and do reduce potential benefits valuable technologies may bring to farmers. Indeed, asking farmers about technology seem to be a first logical step in understanding, yet as the authors have pointed out, has been left out of the equation....
Best,
Jose Falck-Zepeda
Author Response
we would like to thank this reviewer for his appreciation of our paper.
Reviewer 2 Report
I wonder why two key issues of possible reluctance of farmers and of the cotton system are not taken into account, although they might have played an important role in the debate about BT (abandoning):
-farmers might have an aversion against the fact that they cultivate a GM crop (see the video https://www.grain.org/en/article/6092-gmo-cotton-failure-in-burkina-faso-farmers-speak-out where farmers remark that termites do not eat the cotton residues and they conclude that the seeds are poisonous and cannot be fed to animals).
- there was an obvious trade-off between GM variety advantages in spraying and lint quality - thus the ginneries could have opted to pay a lower price for lower quality (which has a cost in internatrional market). In a value chain perspective with all stakeholders having to agree to the choice of technology (in this case for that GM variety) which the authors advocate for, this trade-off should be part of the choice experiment.
Any coment on this (I am aware that this can no longer be incorporated into the experiments and thus the papers, but it seems to be relevant enought to be discussed).
Similarly, I want to remark that while farmers where confronted with a list of variety characteristics, the pre-selection had been made by experts, not farmers themselves. So, the conclusion that "farmers' perferences are mainly shaped by the economic benefits" is pre-shaped by that selection, and other factor (such as the two mentioned above in this remark) simply are not submitted to farmers and thus cannot be taken into account by them and by the researchers in their conclusions. The overall preference for the status quo cantake to a certain extent other reasons into account, but it seems again that a more self-critical discussion and conclusion would be needed. If one wants farmers to really take part in such decisions, the entire cosmos of factors should be taken into account. If the methodology can be extended to include these factors, then it is more convincing.
Finally, I would ask for an assessment in the conclusions whether such method can also be made BEFORE farmers gain experience with the cotton, i.e. a hypothetical situation, which is more relevant at the national level, i.e. to ask farmers BEFORE the decision is taken to start breeding GM crops.

Author Response
We thank the reviewer for all the suggestions and comments.
answer to main comments in the PDF:
-section on limited literature using DCE to assess farmers preferences for GM crops: although indeed the observation that farmers' preferences are more often overlooked (compared to eg consumers) is already from 2008, we did a extensive search for literature using the choice experiment approach to assess this and found really limited papers. and in developing countries it is even more scarce... (the PLOSone paper suggested eg uses a quite different approach looking at adoption form a theory of planned behaviour perspective and the focus is therefore on farmers characteristics determining adoption rather than on crop characteristics) -concerning our sample size : number of farmers interviewed is determined when doing the CE design to be able to do the estimations -notation in equation3 : mistake corrected -difference status quo and current values: for specific attributes the base situation or status quo situation is the same for all farmers belonging to a particular group (bt or non bt) however yield and number of sprays are farm specific ie the point of comparison is individual for each farmer. this is explicitely aknowledged in the modeling framework by including the actual values for these attributes. we have rephrased it in the paper to make this more clear: *Represent the Status Quo level, for “Yield” and “Number of Spray” the actual individual values reported by the farmers are inserted as Status Quo level for the modeling. -on seed distribution and experience we added a footnote : the share of Bt cotton grow to 70 % , but seeds were not distributed systematically to certain GPC ,so when looking at our sample farmers of the ones growing bt some did not have it from the start , of the ones not growing Bt , on average they had also grown it for 2.4 years... interviews seem to indicate that it is mostly a policy by the cotton factories because most GPC were demanding to get Bt seeds on the CL results : indeed the results correspond to common sense and rational behaviour , however the core message is not the sign of the coefficients but the fact that they are significant and thus that these attributes significantly affect the choice of the farmers... because this is not always self evidentThe language and style corrections were also made
comments in letter:
I wonder why two key issues of possible reluctance of farmers and of the cotton system are not taken into account, although they might have played an important role in the debate about BT (abandoning):-farmers might have an aversion against the fact that they cultivate a GM crop (see the video https://www.grain.org/en/article/6092-gmo-cotton-failure-in-burkina-faso-farmers-speak-out where farmers remark that termites do not eat the cotton residues and they conclude that the seeds are poisonous and cannot be fed to animals).
A: when designing the our study we thought about including non-gm options and label the option. however this would steer the study in the direction of a pro and con GM controversy , which was not the objective.
moreover a dislike for GM could be signaled by the farmers by the opt out options alt 4 for the non bt growers and alt 5 for the bt growers
- there was an obvious trade-off between GM variety advantages in spraying and lint quality - thus the ginneries could have opted to pay a lower price for lower quality (which has a cost in internatrional market). In a value chain perspective with all stakeholders having to agree to the choice of technology (in this case for that GM variety) which the authors advocate for, this trade-off should be part of the choice experiment.A: our study takes the perspective of the farmers. the way the value chain was organised farmers where paid per weight ,so for them lint quality was not an issue... with increasing area under bt it became a huge problem for the cotton factories , because they were paying the farmers the same price , but could not get the premium price on the world market anymore.. interviews we had in the sector seem to indicate that it were primarly the cotton factories who put pressure on the government to announce the phase out of bt cotton.
Similarly, I want to remark that while farmers where confronted with a list of variety characteristics, the pre-selection had been made by experts, not farmers themselves. So, the conclusion that "farmers' perferences are mainly shaped by the economic benefits" is pre-shaped by that selection, and other factor (such as the two mentioned above in this remark) simply are not submitted to farmers and thus cannot be taken into account by them and by the researchers in their conclusions. The overall preference for the status quo cantake to a certain extent other reasons into account, but it seems again that a more self-critical discussion and conclusion would be needed. If one wants farmers to really take part in such decisions, the entire cosmos of factors should be taken into account. If the methodology can be extended to include these factors, then it is more convincing.
A: the idea of the paper was, that given the situation with a GM crop in the market for which farmers were not consulted , to which extent does it meet their needs and which point could be improved. Indeed it would also be possible and interesting to assess whether a GM crop at all is something farmers are open to , but that is another starting point ,w hich in the overall framework of our study we found less interesting (certainly for Burkina Faso)
Finally, I would ask for an assessment in the conclusions whether such method can also be made BEFORE farmers gain experience with the cotton, i.e. a hypothetical situation, which is more relevant at the national level, i.e. to ask farmers BEFORE the decision is taken to start breeding GM crops.
A: we have added following sentence in the conclusion :
"A similar methodology could also be used before GM crops are introduced in a country. In this way it could be assessed whether particular crop varieties meet existing needs."
because indeed a strength of the CE method is that you could use it ex ante to study whether certain interventions (like introducing a GM crop) would meet certain needs, be desirable
Reviewer 3 Report
The article is well written and the authors show a detailed and complete knowledge of the agricultural biotechnology issue in Burkina Faso. The statistical analysis seems consistent, and the results are interesting, even if rather obvious as confirmed in the cited bibliography. However, some minor observations must be made.
The title is a little bit misleading, the article doesn’t specifically answer the “What kind of biotechnology do farmers prefer?” question! I would eliminate this part (or reformulate it). I am also quite certain that they would get the same results for any other crop, regardless of whether GM or not.
The choice of three lower price levels 25,000; 17,500; 10,000 FCFA and one higher price of 30,000 FCA seems rather arbitrary, how did you choose these thresholds?
line 305. You wrote “Medium (60 FCFA/Kg)”, but in Table 6, you report the “52” value for Medium farmers group. Please correct
Please check the use of present and past tenses (e.g. line 270When considering the entire sample about 51% of the participants prefer the Status Quo”; line 276Considering the non-Bt growers, approximatively 60% in our sample expressed a willingness to”).
Other minor observations:
-You have used both the CFA and FCFA form, please uniform the style (and maybe it would be useful to use the “CFA franc” wording).
-kg is often written as “Kg” throughout the text, please correct it.
-line 32. Delete one full point after “preferences” (“..”)
-line 159. “a Status Quo”
-line 175. “…the final attributes were selected Often” Yoh have missed a full point.
-line 246. Modify “on of” with “on average of”
Author Response
We thank the reviewer for his appreciation of our work. Below you find an answer to the comments:
The title is a little bit misleading, the article doesn’t specifically answer the “What kind of biotechnology do farmers prefer?” question! I would eliminate this part (or reformulate it). I am also quite certain that they would get the same results for any other crop, regardless of whether GM or not.
A: we have changed the title it now reads:
Farmers’ preferences for cotton cultivation characteristics: A Discrete Choice Experiment in Burkina Faso.
The choice of three lower price levels 25,000; 17,500; 10,000 FCFA and one higher price of 30,000 FCA seems rather arbitrary, how did you choose these thresholds?
A: starting point was the existing price of 27000 FCFA. However in earlier studies is was found that it is a main point of criticism of part of the farmers that this price is too high. we therefore have a range of lower values to see which ones would be acceptable for farmers and which tradeoffs they are willing to make for these lower prices. We also included one higher price to see whether if other attributes (like yield/ seed developer..). would change some farmers might be prepared to pay more than the current price.
line 305. You wrote “Medium (60 FCFA/Kg)”, but in Table 6, you report the “52” value for Medium farmers group. Please correct
A: this was corrected , estimation procedure of WTP was altered but we forgot to change one value in the text
Please check the use of present and past tenses (e.g. line 270When considering the entire sample about 51% of the participants prefer the Status Quo”; line 276Considering the non-Bt growers, approximatively 60% in our sample expressed a willingness to”).
A: we now consistently use the past tense
other minor observations:
-You have used both the CFA and FCFA form, please uniform the style (and maybe it would be useful to use the “CFA franc” wording).
-kg is often written as “Kg” throughout the text, please correct it.
-line 32. Delete one full point after “preferences” (“..”)
-line 159. “a Status Quo”
-line 175. “…the final attributes were selected Often” Yoh have missed a full point.
-line 246. Modify “on of” with “on average of”
A: all these language corrections were done
This manuscript is a resubmission of an earlier submission. The following is a list of the peer review reports and author responses from that submission.
Round 1
Reviewer 1 Report
This article aims to measure the preferences of different types of cotton farmers for alternative growing strategies and technologies. It does so by executing a choice experiment where alternative cotton growing scenarios are presented to 324 cotton farmers. The scenarios are comprised of 5 different attributes (seed prices, yield, number of sprayings, seed provenance and agricultural practices). Based on an analysis of these responses the authors offer some conclusions, including that a majority of non-GM cotton growers are open to changing their current cotton growing strategy.
I do not recommend this article for publication currently given concerns in the methodology of the study, and by extension, the validity of some of its key conclusions. Moreover, it is unclear if enough data is presented to support some of its main conclusions.
The first major area of concern is related to the amount of information given regarding how the study was conceived and executed. A number of important details about the work are entirely absent from the article. Without this information the careful reader does not have the information to judge the validity of the conclusions made by the authors. For example, much more information is needed to understand how the study was conducted, including:
- Location: Where? Why those places? It mentions in the SOFITEX area - and three regions - but this is still a vast geographic space. Locating the villages (on a map) and showing how they are present in different agro-climatic zones (as asserted in the article) will help substantiate this claim. Was agro-climate the only choice for villages? Were other decision criteria used? Since the paper makes claims about all Burkinabe cotton farmers in the conclusions - how is the reader to judge whether these villages and these farmers are representative of Burkinabe cotton farmers generally? What sorts of data / validation can be ushered to prove that claim?
- Time: When was the study conducted? What year? What time of the year? Was this consistent across villages? This is of particular concern since Bt cotton is no longer grown in Burkina Faso. Apparently the study was conducted when it was still being grown, but no year or month of the study is listed. A year is given for the yield component of the study (2015-2016). Is this when the survey was conducted?
- Who: How were particular farmers chosen? It says that a stratified sample was used - but doesn’t say how a master list was obtained, and then how choices of who to administer the survey to were made. Why were only heads of households chosen? Often in cotton growing areas the ‘head of household’ is a figure head, while younger men make particular decisions about agricultural production. Why do the authors think that heads of households are the right decision makers? How can the reader be sure that the sample is representative?
- Execution of the Survey: Who administered the survey? How many different people? How was this process organized? In what language(s)? What if any work was done with survey administrators to ensure the same delivery of the survey across different administrators and languages? Importantly, how were key concepts described to responders? For example, how was the concept of seed provenance described to farmers. The article mentions that most responders don’t have significant formal education - so how is the idea of a public-private seed partnership described? How is the validity / understanding of the question tested? Was the choice experiment tested in a different community with different farmers in order to ascertain its validity (which is standard practice)? If so, what sorts of amendments were made? Answers to these questions are important since they get at the validity of the results and derived conclusions. How is validity checked?
The second major area (which is connected to the first) is related to the types of conclusions drawn from this study and whether they are valid. The authors make a number of conclusions from the study, but it is unclear whether the variable to which they are ascribing a particular outcome is the casual variable. Not enough information is given, or work down, to be clear that this particular variable is actually the key explanatory reason for a particular outcome.
- For example, in the conclusion the authors state (Lines 317 - 319) that conventional farmers are “willing to adopt new GE varieties” - but the data presented here do not support this claim. There are two related concerns here, which hold for other examples in the paper. The first is that it doesn’t appear that work was done to see whether it is correct to say ‘conventional farmers’ are willing to adopt new GE varieties’ - when the population of ‘conventional farmers’ surveyed in this study may have key differences from the ‘Bt group’. For example, they could differ in terms of education, wealth, land size, etc. It doesn’t appear that any work was done to ascertain whether these two groups of farmers were different on key variables - or whether any of these key variables (e.g. education level) were more (or less) explanatory of key decisions. The second concern is that the authors jump from particular characteristics - low cost, low sprayings, etc. - and ascribe that to a “GE varietal” - something that the farmers where never presented with. A cynical way of viewing this would be that conventional farmers are willing to adopt GE varieties that don’t exist (since none are currently available with the characteristics the farmers prefer). Instead the authors should stick to what their data do say: that farmers are willing to adopt new varieties (GE or otherwise) should they provide returns on yield and reduce insecticide use. Jumping to conclusions does a disservice to the empirical work done here in the paper
The paper seems to assume that farmers who are growing Bt or conventional had a choice of which variety to choose. That isn’t necessarily the case. This reviewer’s understanding is that SOFITEX controls seed production (in conjunction with INERA) and then distribute those seeds to GPCs, which are then distributed to individual farmers. Both at the SOFITEX - GPC level, and at the GPC - individual level, choices are constrained by, for example, the availability of different types of seeds by SOFITEX, the preferences of GPC leaders for particular seeds, and then at the individual level, how seeds were distributed amount GPC members. In the least the paper must speak to whether farmers had seed choices, and explain seed choices dynamics. Without such a discussion, the experiments have no relevance for actual farmer choices.
Introduction
The introduction was clearly authored prior to Burkina Faso discontinuing growing of Bt cotton. Curiously the introduction doesn’t mention Burkina Faso’s phase out of Bt cotton due to cotton quality issues (the reader has to wait until section 2.1). This should be moved up to accurately reflect the history of Bt cotton production. The reader should know that farmers no longer have a choice of cotton varietals - Bt is no longer an option. The authors must locate their study in this new context - otherwise the reader will be left wondering why the authors never explicitly addressed this issue.
Specific things to address
The authors cite a 2014 source for the importance of cotton to Burkina Faso. This must be updated to reflect its current status. lines 100-106: This section is confusing and should be rewritten for greater clarity it is puzzling why the authors refer to Burkina Faso’s Bt cotton story as a ‘success’ - given that it is no longer grown there. In the least, any statement of success should be qualified. line 160 refers to “the existing GM variety” Revise to reflect the current absence of GM varieties in Burkina Faso. This entire section includes references as those these GM varieties were still there. It should be stated what year is being referred to - and revise the entire section to reflect this. lines 163-164 refers to non-target resistance to Bt without references. This should be cited. Is this in Burkina Faso or elsewhere? lines 168-170 - This is a key sentence and it is very confusing. It seems to say that farmers have a problem with paying for left over seeds? It is the understanding of this reviewer that the problem has more to do with needing to pay for extra seeds due to climatic hazards. This is a key point that is left out of this ‘farmer choice experiment’ - since farmers must pay the overall cost of seeds (including purchasing supplemental seed) - not just the ‘sticker price’. In the least, the authors should make an acknowledgement that farmers often must purchase additional seed. Seed development: It is unclear how these ‘types’ of seed development were explained to farmers - who only have one reference point for seed purchase. For example, did surveyors describe the GM cotton seed arrangement as ‘private’, ‘public’ or public/private partnership? The authors do not sufficiently explain how yield estimates were made in Table 3. Where/how was the data gathered to make these estimates. Also were there no statistical differences between Bt / non-Bt in the other categories of information gathered (e.g. education). On lines 234-235, the authors mention that “Overall, farmers of this study expressed a positive preference to change their current way to grow cotton by referring to the agricultural practices attribute. “ - But Table 4 indicates that 51% of respondents preferred the status-quo. I see the positive coefficient for ‘Agricultural Practices’ - but it would seem that these two questions / opt out choice are related. In the least it would warrant greater specification if over half want to remain with the status quo. lines 239-242 - the assertion is made that no farmers elected to stop growing cotton and that therefore, this provides support for the conclusion that farmers want to continue to grow Bt cotton. While this reviewer is in accordance with the view that many cotton growers would like to continue with Bt - these two forms of data (electing to continue to grow cotton, and a desire to continue with Bt cotton) are not necessarily connected - and should not be conflated here. Moreover, given that the sample of farmers were only those who were currently growing cotton, and not a larger sample of current and former cotton growers, it is not possible to make a conclusion on whether, as a whole, farmers would like to continue to grow cotton. Other research shows many farmers moving out of cotton production altogether. lines 242-252 appear to make unsubstantiated conclusions from the data - or in the least, stretch what is possible to say according to how the data was collected. The authors state that Bt/non-Bt farmers ‘had opposite preferences’ - this is unclear and potentially misleading. Line 244 states that “51% of the participants prefer to keep their current variety” - But the table presents the choice question as “Alternative 4: I prefer to maintain my current way to grow cotton”. This is much broader than simply the variety. Also, this ‘opt-out’ question is in reference to the choices that are being presented which include yield estimates, number of sprays, seed cost, seed provenance and agricultural practices. So, as this reviewer understands the choice experiment based on what is presented in this paper - the responder could be choosing ‘alternative 4’ based on the alternative options presented to them - none of which they make like, for a variety of reasons. Therefore, it cannot be assumed that their choice is based solely on the ‘variety’ which they are growing. The authors make the unsubstantiated claim that their decisions are based solely on variety - and moreover, that by extension one can make the claim that Bt/nonBt had more or less positive views of the varieties they grow. The claims made in these lines are unsubstantiated by the data. In other words, the authors ascribe variety as the reason for why choices are made - when not enough evidence is provided to make that assertion. not enough information is given to ascertain whether variety is the key characteristic driving these decisions. lines 262-263 revise use of ‘neither’ lines 299-300: This is misleading. Fast uptake is a not simply farmer-driven. Rather, it is driven by seed production dynamics, and what seeds are offered to farmers from cotton companies. lines 303 and 304 are misleading. The authors again seem to conflate the choice experiment. They state that this experiment is important given that farmers don’t necessarily have choice (which wasn’t mentioned until the conclusion. This should be mentioned much earlier in the paper). But the choice experiment is not ostensibly about cotton variety (e.g. Bt / non Bt). lines 319-321: the conclusion includes lines like this: “Regarding the farm size, Small and large farmers groups obtained the highest score by choosing their current situation. “ - which don’t make any sense.Reviewer 2 Report
Dear Authors,
Let me first congratulate you on a quite interesting paper focused on a relevant topic of interest to Sub-Saharan Africa and the potential of GM crops.
I find the paper methods relevant and a straightforward application of choice experiments. Methods are not quite innovative, yet this type of paper is needed to advise policy and decision making based on robust evidence.
Apart from an English style and spelling check, there are two issues that I believe require minor revisions.
One, is a conceptual item for clarification. This relates to Page 2 paragraph lines 55-64. Here you discuss smallholder cotton farmers in Burkina issues. My issue is that taking into account the top-down nature of industrial organization in Burkina, how can we differentiate that what you found in the analysis truly reflects farmer preferences and eventual adoption vis-a-vis what variety the cooperative or seed association may have given them as the "choice". I realize this is not a adoption study per se, just need and expanded clarification of the issues related to farmers preferences and adoption. A few sentences will suffice.
Second, is that after quite an interesting discussion of results, seems to be a bit anti climatic not to discuss even briefly, policy implications of your results. I realize this is not a policy journal, so an extensive discussion is not needed. Yet, even a sketch of potential policy implications in paragraph or two, will bring home the message to readers. If you throw in implications for agronomy and plant improvement scientists, even better.
Once again, thanks for a quite nice paper.